# Effect of Modification with Helium Atmospheric-Pressure Plasma and Deep-Ultraviolet Light on Adhesive Shear Strength of Fiber-Reinforced Poly(ether-ether-ketone) Polymer

**DOI:** 10.3390/jfb11020027

**Published:** 2020-05-01

**Authors:** Seigo Okawa, Norimasa Taka, Yujin Aoyagi

**Affiliations:** 1Division of Biomimetics, Faculty of Dentistry & Graduate School of Medical and Dental Sciences, Niigata University, 2-5274, Gakkocho-dori, Chuo-ku, Niigata 951-8514, Japan; 2Division of Bio-Prosthodontics, Faculty of Dentistry & Graduate School of Medical and Dental Sciences, Niigata University, 2-5274, Gakkocho-dori, Chuo-ku, Niigata 951-8514, Japan; taka@dent.niigata-u.ac.jp (N.T.); aoyagiy@dent.niigata-u.ac.jp (Y.A.)

**Keywords:** fiber-reinforced poly(ether-ether-ketone), surface modification, shear bond strength, surface analysis

## Abstract

We investigated the effect of helium atmospheric-pressure plasma (PL) and deep-ultraviolet (UV) light treatments on the adhesive properties of fiber-reinforced poly(ether-ether-ketone) polymer (PEEK). PEEK disks reinforced with carbon (CPEEK) or glass (GPEEK) fibers were polished, modified with PL and UV for 60 s, and the surface energy was calculated by measuring the contact angles. The disk surfaces were analyzed by X-ray photoemission spectroscopy. Shear bond strength testing was performed using a universal testing machine, and the fracture surfaces were observed by electron probe microanalyzer. Data were analyzed with one and two-way ANOVA and Tukey’s post-hoc test (*p* < 0.05). The surface energies were increased by the modifications, which created OH functional groups on the surfaces. The bond strengths of CPEEK were increased by PL, and those of GPEEK were increased by PL and UV, owing to chemical bonding at the interface.

## 1. Introduction

The biocompatible polymer poly(ether-ether-ketone) (PEEK) has been used in medical applications, including trauma, orthopedic, and spinal implants [1], and in dental implants, dental CAD/CAM blocks, and bone plates [2,3]. PEEK is safe and stable in the body and has excellent chemical and mechanical properties compared with the conventional polymer poly(methyl methacrylate)(PMMA) (Table 1) [4]. However, there are some aesthetic drawbacks that limit the use of PEEK as dental prosthesis. Moreover, the elastic modulus of PEEK is slightly lower than that of bone, although the modulus of fiber-reinforced PEEK is similar to that of bone [5]. PEEK and fiber-reinforced PEEK are chemically inert and therefore possess poor adhesive properties. Hence, in order to facilitate their widespread use as biomaterials, they require improvement of the adhesive properties.

In previous studies [6,7,8,9,10,11,12,13,14,15,16,17,18,19,20,21,22,23], the adhesion of PEEK can be improved by different physical and chemical surface modifications and treatments, including sandblasting [6,7,8,9,10], acid treatment [8,9,10,11,12,13], laser beam emission [14,15,16], ultraviolet (UV) light irradiation [17], and plasma treatment [18,19,20,21,22,23]. These studies showed that the bond strengths of PEEK and fiber-reinforced PEEKs increase following their surface modification. Zhou et al. reported that the shear bond strength values of sandblasted and acid-etched PEEKs were 5.3 ± 0.6 and 8.7 ± 0.2 MPa, respectively [10]. Thus, the authors concluded that sandblasting and chemical treatment produced highly porous surfaces. Mechanical coupling could be achieved by their treatments. On the other hand, Laurens et al. described that pulsed excimer laser enhanced the adhesive bonding properties of PEEK films [16]. Moreover, Zhang et al. showed that plasma treatment of the PEEK films enhanced their bonding strength [19]. These modifications allowed chemical bonding with PEEK.

In regard to the fiber-reinforced PEEK, however, the relationship between the chemical bonding mechanism and the surface modifications is not fully understood. In this study, the surfaces of carbon fiber-reinforced PEEK (CPEEK) and glass fiber-reinforced PEEK (GPEEK) with elastic modulus similar to that of bone were modified by helium atmospheric-pressure plasma (PL) and UV light with a wavelength of 172 nm. This study investigates the effect of modification on the surface adhesive properties and adhesive strength of the fiber-reinforced PEEKs.

## 2. Materials and Methods

### 2.1. Specimen Preparation

CPEEK rod (Ketron CA30 PEEK, Quadrant Polypenco Japan, Ltd., Tokyo, Japan) and GPEEK rod (Ketron GF30 PEEK, Quadrant Polypenco Japan, Ltd.) were cut into disks approximately 5 mm thick with a cutting unit. Half of the disks were embedded in thermal curing resin to prepare adherend specimens. The end surfaces of the embedded disks were polished with 1000-grit SiC waterproof paper under running tap water. The disks were cleaned by ultrasonication for 10 min in distilled water, and then dried at room temperature.

### 2.2. Surface Modification

PL (Piezobrush PZ2, Relyon Plasma, Regensburg, Germany) and a 172 nm UV light (Min-Excimer SUS713, Ushio Denki, Tokyo, Japan) were used to modify the surface of the fiber-reinforced PEEKs. The surfaces of the adherend and cylindrical PEEK specimens were irradiated with PL and UV light emission for 60 s.

### 2.3. Measurement of Contact Angle and Evaluation of Surface Energy

Contact angle measurements were performed by the sessile drop method. Two testing liquids, distilled water and formamide (98.5% purity, FUJIFILM Wako Pure Chemical Co., Tokyo, Japan) with well-known polar and dispersive components of surface energy were used. Drops of testing liquid (5.0 μL) were deposited from a micro syringe onto the specimen surfaces at 20 ± 1 °C. Each contact angle was acquired from images captured by a charge coupled device (CCD) camera when observable motion had ceased. The polar and dispersive components of surface energies of the modified specimens were calculated from the experimental value using method described by Jha et al. [22]. Three measurements on each specimen were done separately.

### 2.4. Surface Analysis

The PEEK specimens were analyzed by X-ray photoemission spectroscopy (XPS; Quantum 2000, ULVAC, Chigasaki, Japan). Incident monochromated X-rays from the Al target (100 W) were focused on a 100-μm-diameter area. XPS spectra analysis was carried out with MultiPak software (ULVAC). To analyze the special functional groups formed by the surface modifications qualitatively, selective chemical derivatization techniques were performed as reported elsewhere [24,25]. To identify OH functional groups, trifluoroacetic anhydride (TFAA; Tokyo Chemical Industry, Co., Ltd., Tokyo, Japan) was poured into the outer vessel of a double-walled glass vessel, the modified specimen was put into the inner vessel, and a glass cover was placed over the top of the vessel. The specimen was exposed to TFAA vapor at room temperature for 24 h. To identify COOH functional groups, a 9:3:4 (*v*/*v*) mixture of 2,2,2-trifluoroethanol (Tokyo Chemical Industry, Co., Ltd.), N,N′-di-tert-butylcarbodiimide (Tokyo Chemical Industry, Co., Ltd.), and pyridine (FUJIFILM Wako Pure Chemical, Co., Tokyo, Japan) was placed in the outer vessel of a double-walled glass vessel and the modified specimen was placed in the inner vessel and exposed to the vapor at room temperature for 24 h. Initial XPS measurements were carried out on the C1s peaks. Generally, polymers are non-conductive, however, the X-ray excitation mechanism and the ejection of the photoelectrons from the specimen surface causes the buildup of a positive charge on the surface, which shifts spectra by a few electron volts toward higher binding energies. Therefore, a neutralizer attached to the XPS apparatus was used for charge compensation while XPS spectra were measured. The high-resolution C1s peak spectrum of the chemically modified specimens contained contributions from carbon atoms in various chemical environments. Peak fitting by MultiPak software was used to identify the grafted components.

### 2.5. Bonding Procedure

A punching seal with a diameter of 6.0 mm was placed on the modified adherend specimen to keep the adhesive area constant. The specimen was bonded with the cylindrical specimen by using a methyl methacrylate (MMA) adhesive resin (Super-Bond C&B, Sun Medical Co., Ltd., Moriyama, Japan) according to the manufacturer’s instructions (n = 10). After bonding, the specimens were left for 1 h under atmospheric conditions and then stored in distilled water at 37 ± 1 °C for 24 h.

### 2.6. Compressive Shear Bond Strength Test

The specimen was placed on the compressive shear bond strength test apparatus and a universal testing machine (Autograph-1000E, Shimadzu, Kyoto, Japan) was operated at a crosshead speed of 1.0 mm/min. The shear bond strength was calculated by dividing the stress on failure by the area.

### 2.7. Observation of Fractured Surfaces

The fractured surfaces of the specimens were coated with AgPd and secondary electron (SE) images were acquired using an X-ray probe microanalyzer (EPMA-1610, Shimadzu).

### 2.8. Statistical Analysis

Statistical differences were analyzed with one- and two-way ANOVA and Tukey’s post hoc tests (*p* < 0.05) using Microsoft Excel.

## 3. Results

The contact angle measurement values on the modified specimens are listed in Table 2. It is observed that surface modification of PL and UV results in significant decrease in contact angle compared to the contact angle of the control.

The surface energies are shown in Figure 1. Statistically significant differences between the surface energy of PL-modified CPEEK and the control were found. The dispersive component decreased, and the polar component was more than 4 times higher in the PL-modified specimens compared with the control. However, there was no statistically significant difference between the control and CPEEK modified by UV. The polar and dispersive components of the surface energy were similar in the CPEEK specimens modified by UV. In contrast, the polar component of the surface energy of the GPEEKs significantly increased, and the surface energies of the GPEEKs modified by PL and by UV were significantly higher than that of the control.

The O/C atomic ratios obtained by XPS are listed in Table 3. The O/C ratios of the PL- and UV-modified specimens were higher than that of the control because surface modifications increased the surface oxygen content. The oxygen contents for the UV-modified specimens were higher than for the PL-modified specimens.

Figure 2 shows the fitted C1s spectra of the modified specimens. Table 4 shows the XPS reference table containing the C1s binding energies for aliphatic species. The spectra contained C-H and C-C (~285.0 eV), C=O (287.3 eV), and π-π * (291.5–292.0 eV) bands. The characteristic π-π * shake-up satellite arose from the resonance of the aromatic rings of PEEK. An additional peak corresponding to O–C=O, COOH, and OH (~289.0 eV) was present in the spectra for the specimens modified by PL and UV light.

The peak area at around 289.0 eV was larger for the UV-modified specimens than the PL-modified specimens.

Selective chemical derivatization was performed to identify the specific functional groups, and the XPS spectra of the derivatized specimens are presented in Figure 3. OH groups on the modified specimen surfaces were indicated by C1s peaks at 293.2 eV on the PL- and UV-modified specimens derivatized with TFAA. No C1s peak attributed to CF_3_ at 293.2 eV was observed on the control. The presence of COOH groups was indicated by the C1s peaks at 293.2 eV for PL- and UV-modified CPEEK, although these C1s peaks were not observed for PL- and UV-modified GPEEK.

A box plot of the compressive shear bond strengths is shown in Figure 4. The bond strengths of the PL-modified CPEEK and GPEEK specimens were significantly higher than those of the control (unmodified specimen). However, UV-modified CPEEK showed no statistically significant difference from the control. The bond strength of PL-modified CPEEK was significantly higher than that of UV-modified CPEEK. In contrast, the bond strengths of PL- and UV-modified GPEEK were similar.

SE images of the polished specimens and the fracture surfaces are shown in Figure 5. The reinforcing fibers in CPEEK and GPEEK were 3–4 and 11–13 μm in diameter, respectively. The SE images of the fracture surface showed that the adhesive resin was not bonded to the fibers in both specimens. In view of the form of original surface (as polished), in the CPEEK control, some pieces of adhesive resin were observed in the space around the fibers, whereas there was no resin in these spaces in the GPEEK control. Moreover, a flat surface was observed at the interface. In addition, in PL-modified CPEEK and GPEEK, some pieces of broken adhesive resin were visible along the polishing streaks. In contrast, in UV-modified CPEEK, the fracture properties were similar to those of the CPEEK control, and some pieces of adhesive resin were observed in the space around the fibers but not around the polishing streaks. In UV-modified GPEEK, there were wave-like traces of the resin fracture on the interface.

## 4. Discussion

To use the biocompatible polymer PEEK as a medical material, problems with its bonding properties and bond strength must be solved. To overcome these problems, we modified fiber-reinforced PEEKs, which have elastic moduli similar to that of bone modulus, by PL and 172 nm UV light to improve their bond strength.

The surface energies of PL-modified CPEEK and GPEEK were significantly higher than those of the controls, indicating that PL created hydrophilic surfaces. XPS spectra of the derivatized surface structures of PL- and UV-modified CPEEK and GPEEK contained CF_3_ or C-F peaks, corresponding to species formed by the reaction of the TFAA derivatization agents with isolated OH species introduced by the modifications on these specimens. In contrast, COOH species were formed on the PL- or UV-modified CPEEK specimens, although they were not formed on the PL- or UV-modified GPEEK specimens because no C1s peak at 293.4 eV was observed. The results confirmed the presence of OH and COOH functional groups. Zhang et al. [19] reported that C-O and COO functional groups were formed on the surface of PEEK modified by plasma. Moreover, Iqbal et al. [20] found the same functional groups on PEEK modified by PL by XPS analysis. Our findings agree well with these results.

The ions and electrons created from PL and UV light energy could create these functional groups through the chain scission of C-O-C in ether groups and C=O-O in ketone groups in PEEK. Gonzalez et al. [26] reported that the O atoms generated in PL oxidize and open the aromatic rings on the polymer chains. The energy of 172 nm UV light is 694.9 kJ/mol, and the bond dissociation energy of the ester groups in PEEK is 347.4 kJ/mol [27]. Therefore, the ester bonds easily underwent scission, oxygen in the air was dissociated into oxygen radicals, and ozone was formed. The surfaces of the specimens were rich in OH and COOH functional groups owing to the attack of these chemical species, consistent with the increase in O/C ratio.

Niu et al. [28] reported that UV light caused the chain scission of C–O–C in ether groups and C=O-O in ketone groups, and -OH groups were formed in different positions. Moreover, Riveiro et al. [15] reported that irradiation with a 355 nm UV laser was the most suitable for modifying the PEEK surface, suggesting that UV light with a wavelength of around 355 nm may improve the surface modification of CPEEK and GPEEK.

The shear bond strengths are shown as a box plot (Figure 4). The shear bond strengths of both specimens modified by PL were significantly higher than those of the control. However, there was no statistically significant difference in the shear bond strength between UV-modified CPEEK and the control. Based on the XPS results, OH and COOH functional groups were formed on the CPEEK surface by UV irradiation. The deconvolution and curve fitting of the C1s XPS spectra of the UV-modified GPEEK and CPEEK specimens were identical. However, there was no statistically significant difference between the surface energy of UV-modified CPEEK and the control. The fracture surface properties of the UV-modified CPEEK and the control were similar, and polishing streaks were observed on the interface of both specimens. These results indicate that few chemical bonds were formed on the surface of UV-modified CPEEK. According to the O/C ratio after UV irradiation, the modified surface had high oxygen content because the ozone and active oxygen formed by the UV light irradiation reacted with the CPEEK surface. Therefore, the modified surface contained not only abundant OH functional groups but also would have a new compound with OH functional groups. In fact, Niu et al. [28] reported that the short wavelengths (300–380 nm) provided by UV light were completely absorbed by the PEEK polymer, and low-molecular-weight products were formed. Because CPEEK contains carbon fibers, the ozone and active oxygen formed by the UV light irradiation would readily react with the carbon fibers to form carbon and CO_2_. These materials would easily form low-molecular-weight products. The shear bond strength of the PL-modified CPEEK was higher than that of the UV-modified CPEEK because PL did not produce low-molecular-weight products.

The reaction described above would also occur on the specimens modified by PL or UV light. Therefore, PL and UV treatment are suitable for modifying GPEEK, and UV treatment is suitable for modifying CPEEK.

According to the SE images, the glass fibers were thick. Because the adhesive resin does not chemically bond to the reinforced fibers, the real bonding area of CPEEK was higher than that of GPEEK, resulting in the lower bond strength of GPEEK. This was consistent with the bond strength results. In contrast, mechanical fastening occurred on the CPEEK control. Some pieces of the resin were stuck in the polishing streaks on the fracture surface of the CPEEK control, demonstrating the excellent penetration of the MMA-based adhesive resin. Some pieces of the adhesive resin were attached to the fracture surfaces of PL-modified CPEEK and GPEEK and UV-modified GPEEK. Thus, cohesive failure may occur at each interface. Consequently, mechanical and chemical bonds occurred between the adhesive resin and CPEEK and GPEEK, and a strong bond was formed at the interface.

## 5. Conclusions

The adhesive properties of PL- and UV-modified CPEEK and GPEEK were investigated. These treatments introduced OH functional groups onto the surfaces of CPEEK and GPEEK. The bond strengths of CPEEK were increased by PL, and those of GPEEK were increased by both PL and UV light. Mechanical fastening occurred, and chemical bonds were formed, which was indicated by cohesive failure at the modified interface.

## Figures and Tables

**Figure 1 jfb-11-00027-f001:**
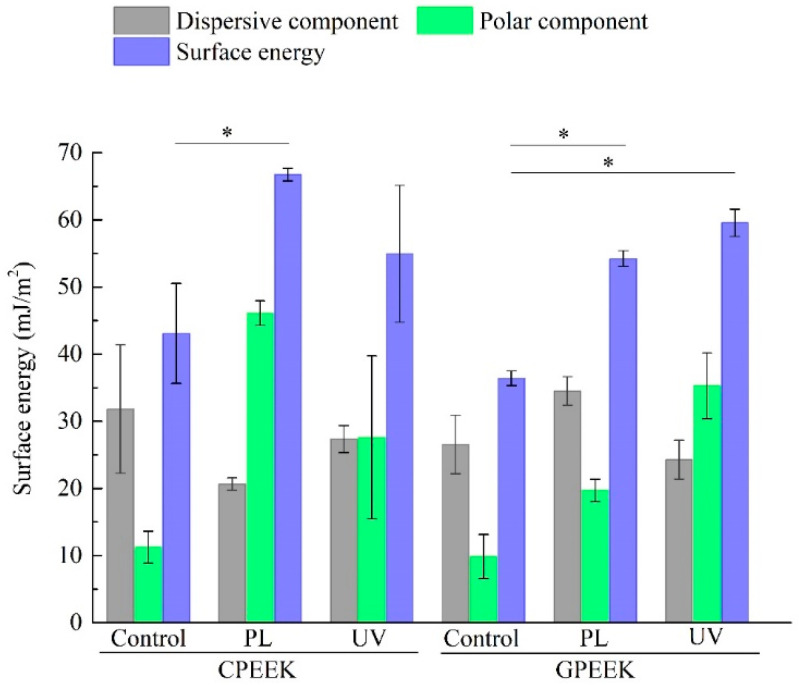
Surface energies of the modified specimens. CPEEK: carbon fiber-reinforced PEEK, GPEEK: glass fiber-reinforced PEEK, Control: unmodified specimen, PL: helium atmospheric-pressure plasma, UV: deep-ultraviolet light. Asterisks indicate *p* < 0.05 compared to the specimens. Bar denotes standard deviation.

**Figure 2 jfb-11-00027-f002:**
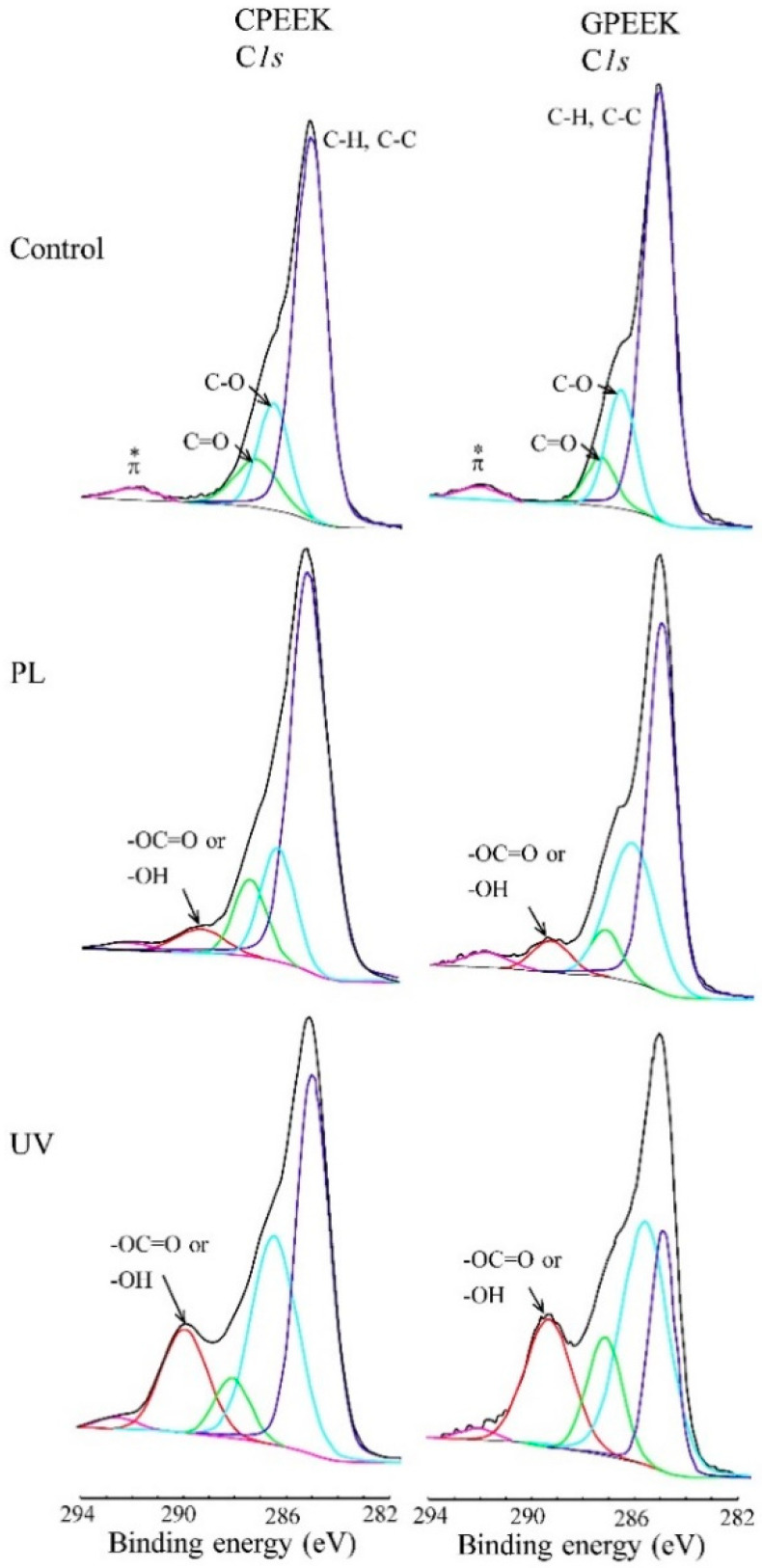
C1s fitted XPS peak spectra for the modified specimens. CPEEK: carbon fiber-reinforced PEEK, GPEEK: glass fiber-reinforced PEEK, Control: unmodified specimen, PL: helium atmospheric-pressure plasma, UV: deep-ultraviolet light.

**Figure 3 jfb-11-00027-f003:**
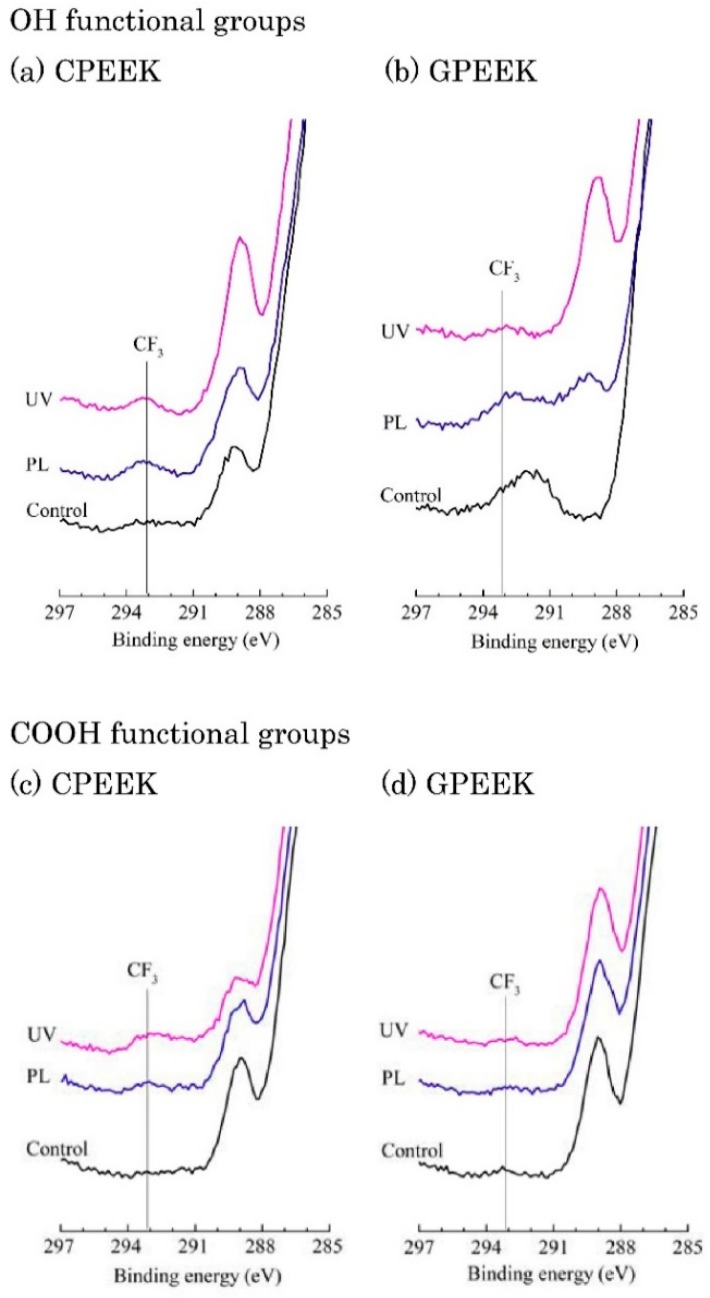
C1s XPS spectra of the modified specimens after derivatization. Labelled C1s core-level fits using peak for CF_3_. CPEEK: carbon fiber-reinforced PEEK, GPEEK: glass fiber-reinforced PEEK, Control: unmodified specimen, PL: helium atmospheric-pressure plasma, UV: deep-ultraviolet light.

**Figure 4 jfb-11-00027-f004:**
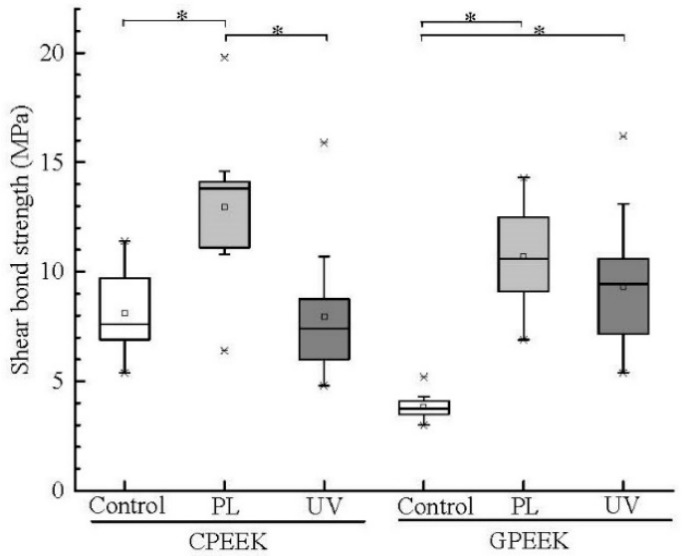
Box plot of the compressive shear bond strengths. CPEEK: carbon fiber-reinforced PEEK, GPEEK: glass fiber-reinforced PEEK, Control: unmodified specimen, PL: helium atmospheric-pressure plasma, UV: deep-ultraviolet light. Asterisks indicate *p* < 0.05 compared to the specimens. X marks denote outliers. Small square in the box indicates the mean value.

**Figure 5 jfb-11-00027-f005:**
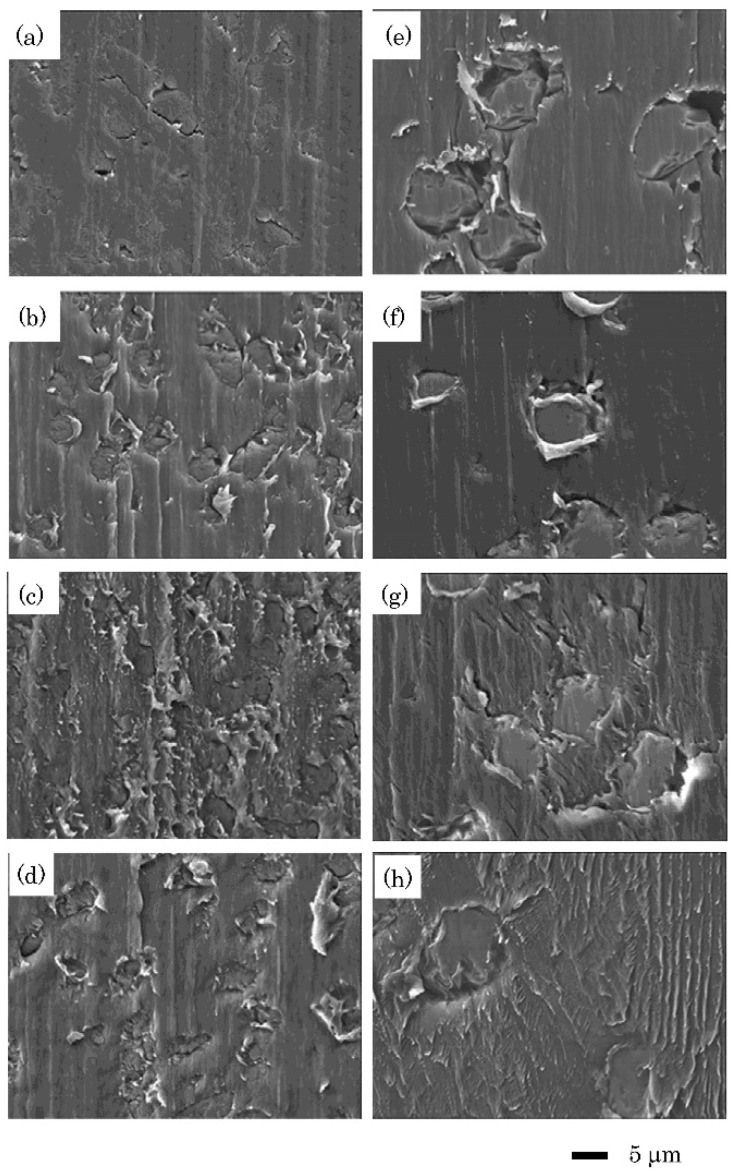
SE images of the fractured specimens: (**a**) CPEEK as polished, (**b**) CPEEK control, (**c**) CPEEK PL, (**d**) CPEEK UV, (**e**) GPEEK as polished, (**f**) GPEEK control, (**g**) GPEEK PL, (**h**) GPEEK UV. CPEEK: carbon fiber-reinforced PEEK, GPEEK: glass fiber-reinforced PEEK, Control: unmodified specimen, PL: helium atmospheric-pressure plasma, UV: deep-ultraviolet light.

**Table 1 jfb-11-00027-t001:** Mechanical and physical properties. PMMA: poly(methyl methacrylate), PEEK: poly(ether-ether-ketone), CPEEK: carbon fiber-reinforced PEEK, GPEEK: glass fiber-reinforced PEEK. According to the information provided by the manufacturer.

Mechanical and Physical Properties	PMMA	PEEK	CPEEK	GPEEK
Specific gravity	1.18	1.31	1.41	1.51
Tensile strength (MPa)	55–76	110	131	97
Tensile modulus of elasticity (GPa)	2.4–3.4	4.3	7.6	6.9
Tensile elongation (at break) (%)	2	40	5	2
Flexural strength (MPa)	83–117	172	178	159
Flexural modulus of elasticity (Gpa)	2.4–3.4	4.1	8.6	6.9
Shear strength (MPa)	-	55	103	97
Water absorption immersion 24 h (% by wt)	0.3	0.1	0.06	0.1

CPEEK: 30% Glass fiber filled PEEK, GPEEK: 30% Glass filled PEEK.

**Table 2 jfb-11-00027-t002:** Contact angle measurement values on the specimens. CPEEK: carbon fiber-reinforced PEEK, GPEEK: glass fiber-reinforced PEEK, Control: unmodified specimen, PL: helium atmospheric-pressure plasma, UV: deep-ultraviolet light.

Liquid	Modifications	Specimens	
		CPEEK	GPEEK
		Mean (SD)	Mean (SD)
Distilled water	Control	88.6 (0.9)	74.4 (3.0)
	PL	37.2 (2.6)	37.3 (4.2)
	UV	23.8 (2.0)	51.4 (1.6)
Formamide	Control	55.3 (2.8)	55.1 (1.2)
	PL	11.2 (0.8)	19.4 (0.6)
	UV	8.6 (0.7)	21.8 (3.6)

Unit: °.

**Table 3 jfb-11-00027-t003:** O/C atomic ratios of the modified specimens. CPEEK: carbon fiber-reinforced PEEK, GPEEK: glass fiber-reinforced PEEK, Control: unmodified specimen, PL: helium atmospheric-pressure plasma, UV: deep-ultraviolet light.

Specimen	Control	PL	UV
CPEEK	0.20	0.44	0.61
GPEEK	0.26	0.38	0.61

**Table 4 jfb-11-00027-t004:** Functional groups and their XPS binding energies. CPEEK: carbon fiber-reinforced PEEK, GPEEK: glass fiber-reinforced PEEK, Control: unmodified specimen, PL: helium atmospheric-pressure plasma, UV: deep-ultraviolet light.

	Binding Energy (eV)			References		
	Control	PL	UV	[19]	[21]	[23]
C-H/C-C	285.1	285.0	285.0	285	285.0	285.0
C-O-C	286.5	286.3	286.3	286.4	286.3	286.4
C=O	287.3	287.3	287.3	288	287.4	287.2
COOH or COH	289.0	289.0	289.2	289	288.7	
π-π *	292.0	291.7	291.5	291.79	291.7	291.8

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
