# Peer review of "Effect of Modification with Helium Atmospheric-Pressure Plasma and Deep-Ultraviolet Light on Adhesive Shear Strength of Fiber-Reinforced Poly(ether-ether-ketone) Polymer"

_jfb, 2020, doi:10.3390/jfb11020027_

Round 1

Reviewer 1 Report

This manuscript titled: “Adhesive properties of fiber-reinforced poly(ether-ether-ketone) polymer modified with helium atmospheric-pressure plasma and deep-ultraviolet light”, it describes the effect of two different dry surface treatments in two different PEEK composites, one with carbon fibers and the other with glass fibers, for the adhesion with an adhesive resin. In my opinion, the results are well described and discussed, and though, the results are foreseeable and not very impressive in a scientific point of view, are interesting for practical applications. Then, my recommendation is to publish this manuscript in Journal of Functional Materials after a minor revision of these few points:

  1. In general, introduction and experimental part are a little bit poor, more details about the importance of these treatments for some applications could enrich the introduction.
  2. In the case of experimental part, I miss more details. For example, how the dispersive component and polar component were calculated, or at least to add a reference where the methodology was described. I guess they were calculated from the sessile drop method. For these experiments I proposed to author to add the results of contact angles, because they are much more common and more people can understand the effect of treatment with these contact angle values.
  3. Authors could add the nature of resin in the experimental part. In the discussion, they mentioned that the resin is MMA, this information helps to understand the important of the treatments.
  4. Finally, I miss discussion about the hydrophobic recovery. It is well reported that after a plasma treatment the contact angle (for water) decreases a lot, but after few hours or days the contact angle increases again because the material reduces the surface energy, migrating the polar groups from the surface to the inner part. Have authors studied this effect in PEEK composites or they know something about its importance in these composites?

Author Response

We wish to express our appreciation to the reviewers for their insightful comments on our paper. The comments have helped us significantly improve the paper.

  1. In accordance with the reviewer's comment, we have added introduction to new text.
  2. We agree that additional information on the results of contact angles as the reviewer suggested would be valuable. We have added the following to this section. On the basis of the report by Jha S [22], we decided the surface energy. 
  3. We have added the nature of resin in Table 1.
  4. We have no data of the contact angles after few hours or days. However, we compared the bond strength of the specimen modified immediately with that of after 3 days. There was no statistically difference between both specimens.

Thank you once again for your valuable comments and suggestions. We are hopeful that our supplementary analyses and revised focus helps to improve your opinion of work.

Reviewer 2 Report

The paper by Okawa Seigo et al. describes the effect of helium atmospheric-pressure plasma and deep UV light treatments on the adhesive properties of PEEK fiber-reinforced with carbon or glass. The surface properties were investigated by the contact angle measurements, surface free energy calculation (Owens-Wendt equation), XPS, shear bond strength and fracture surfaces testing. The presence of -OH and -COOH functional groups on the surface as a result of modification was confirmed.

In my opinion, this paper is relevant to the field of the journal. The obtained results are interesting and concisely presented. The originality and scientific quality of this manuscript meet the acceptable standards. These along with careful explanations, make the paper publishable. I propose to accept this paper for publication after revision according to the following remarks:

Page 2, line 58: Why two polar liquids (water and formamide) were used for contact angle measurements? To calculate the polar and dispersive components of the surface free energy by the Owens-Wendt approach, both polar and apolar liquids are needed. The equation should be provided. There is lack of information on the droplet size, volume.

Page 2, line 58: The purity of water used for measuring should be given.

Page 3, Fig. 1: For clarity, the error deviations should be visible not only as positive but also as negative ones, such as it is presented for the dispersive component of CPEEK modified with UV. The explanation of symbols, stars and lines, should be placed in the Figure caption or in the text.

Page 6, Fig. 4: The explanation of symbols (stars, stretches) is recommended.  

Page 7, lines 149-155: The description of how to identify the resin in SE images should be improved.

Page 7, line 183: The reference [15] should be black.

Page 8, line 214: The shortcut MMA should be explained.

Pages 9 and 10: Reference list should be improved. In some cases there are full names of the journals while in others their abbreviations. Also pay attention to abbreviations, they are not always correct.

Author Response

Thank you for all of your detailed comments and suggestions. We found them quite useful as we approached our revision.

  1. Page 2, line 58:

We have added the following to this section. On the basis of the report by Jha S [22], we decided the surface energy. The drop was withdrawn at 5 mL by using a micro syringe.

  1. Page 2, line 58:

We have added the following to this section. We used MilliQ water and formamide to measure the contact angles.

  1. Page 3, Fig. 1:

We have added the explanation of symbols, stars and lines in the figure caption.

  1. Page 6, Fig. 4:

We have added the explanation of symbols in the figure caption.

  1. Page 7, lines 149-155:

We have added the following to this section. We identified the resin in SE image from the form of original surface (as polished).

  1. Page 7, lines 183:

The reviewer’s comment is correct.

  1. Page 8, lines 214:

We have added the explanation of the shortcut MMA in materials and methods section.

  1. Pages 9 and 10:

The reviewer’s comment is correct.

Again, we appreciate all of your insightful comments. We worked hard to be responsive to them. Thank you for taking the time and energy to help us improve the paper.
